# MicroRNA as a Potential Therapeutic Molecule in Cancer

**DOI:** 10.3390/cells11061008

**Published:** 2022-03-16

**Authors:** Joanna Szczepanek, Monika Skorupa, Andrzej Tretyn

**Affiliations:** 1Centre for Modern Interdisciplinary Technologies, Nicolaus Copernicus University, Ul. Wilenska 4, 87-100 Torun, Poland; monika_skorupa@umk.pl; 2Faculty of Biological and Veterinary Sciences, Nicolaus Copernicus University, Ul. Lwowska 1, 87-100 Torun, Poland; prat@umk.pl

**Keywords:** miRNA, replacement therapy, drug resistance, oncomiRs, tumor suppressor, metastamiRs, miRNA delivery systems, miRNA inhibition therapy

## Abstract

Small noncoding RNAs, as post-translational regulators of many target genes, are not only markers of neoplastic disease initiation and progression, but also markers of response to anticancer therapy. Hundreds of miRNAs have been identified as biomarkers of drug resistance, and many have demonstrated the potential to sensitize cancer cells to therapy. Their properties of modulating the response of cells to therapy have made them a promising target for overcoming drug resistance. Several methods have been developed for the delivery of miRNAs to cancer cells, including introducing synthetic miRNA mimics, DNA plasmids containing miRNAs, and small molecules that epigenetically alter endogenous miRNA expression. The results of studies in animal models and preclinical studies for solid cancers and hematological malignancies have confirmed the effectiveness of treatment protocols using microRNA. Nevertheless, the use of miRNAs in anticancer therapy is not without limitations, including the development of a stable nanoconstruct, delivery method choices, and biodistribution. The aim of this review was to summarize the role of miRNAs in cancer treatment and to present new therapeutic concepts for these molecules. Supporting anticancer therapy with microRNA molecules has been verified in numerous clinical trials, which shows great potential in the treatment of cancer.

## 1. Introduction

MicroRNAs are an abundant class of endogenous small RNA molecules (18–22 nucleotides in length) that are noncoding post-transcriptional modulators of gene expression [1,2,3]. The human genome produces nearly 2000 miRNAs. Approximately 1900 precursors and 2600 mature human miRNA sequences are indexed in the miRBase database (http://www.mirbase.org/, accessed on 25 October 2021) [4]. The majority of miRNAs still await discovery, but some of them may be cancer-specific markers. They regulate gene expression by suppressing mRNA translation, mRNA cleavage, and mRNA decay initiated by miRNA-guided rapid deadenylation and reducing mRNA stability [5,6]. Each miRNA can control hundreds of target genes, so identifying key miRNA targets for cancer research is an important aspect.

Research on identifying single or sets of miRNAs, as regulators of cell proliferation and apoptosis processes, is currently one of the most promising areas of research. The role of these molecules is varied (Table 1). Some miRs exert negative control over the expression of many oncoproteins in normal cells, and therefore their deregulation is believed to be a significant mechanism underlying the development and progression of cancer. Due to their role in the process, there are three categories of microRNA: oncogenic miRNAs (oncomiRs), tumor suppressor miRNAs, and metastatic miRNAs (metastamiRs) (Table 1). The consequence of the overexpression of oncomiRs is the initiation, development, progression, and invasion of neoplastic disease. The miRNA classification is not clear, because many of the same miRNAs (e.g., miR-7 [7,8,9], miR-125b [10,11], miR-155 [12,13,14], and miR-30b/30d [15,16]) function as oncogenes and also as tumor suppressor genes [3,17]. This property, however, increases the therapeutic potential of these molecules.

In many studies, it has been confirmed that microRNAs play important roles in the pathogenesis of human cancers. Identification of the expression patterns associated with specific tumor cell phenotypes may open new possibilities for the early diagnosis and therapy of cancer. Specific miRs expression patterns were identified for lung [31,32,33,34], breast [18,24,35,36,37], brain [38,39,40,41,42], liver [43,44,45,46,47,48], and colorectal cancer [49,50,51,52,53,54], and leukemia [2,55,56,57,58]. MiRNA signatures can be useful in developing new cancer prevention strategies and also new cancer therapy options. Although the role of miRNA in the pathogenesis of human cancers is proven, the exact mechanisms of regulation of the multiple stages of pathogenesis (initiation, promotion, malignant conversion, progression, and metastasis) are still unknown [5]. MicroRNAs are interesting biomarkers for several reasons, but the most important of them are the different expression patterns associated with the type of cancer, their remarkable stability, and their easy and noninvasive identification due to the presence in the blood and other body fluids [59].

## 2. MicroRNA’s Correlations with Therapy

The association between changes in the level of microRNA expression and therapy has been demonstrated for several years, including the relationship with drug resistance and modulation of the response to cancer therapy, as well as the development of new cancer therapy regimens and new targeted drugs. However, the introduction of miRNAs into therapy requires a holistic approach that takes into account both the multiplicity of miRs in cancer cells and the interactions of miRNAs with the immune system, tumor stromal cells, cancer therapies, and other factors extrinsic to the cancer cells themselves [17].

### 2.1. MicroRNA and Drug Resistance

The development of resistance to anticancer drugs is one of the most serious causes of therapy failure. Over the past few decades, this complex and multifactorial process of cancer cells acquiring resistance has been intensively studied, leading to the identification of various genetic biomarkers and mechanisms responsible for the phenomenon. Nevertheless, the interrelationships between specific cancer subtypes as well as the specific biomarkers and disturbances in the molecular pathways are still not fully understood and described. In the context of this research, in recent years, microRNAs have become an object of interest because of their ability to regulate the expression of genes involved in cell responses to drugs. Hundreds of miRNAs have been identified as biomarkers of drug resistance, and many have demonstrated the potential to sensitize cancer cells to therapy. Based on an analysis of their expression signatures, microRNAs playing a role in regulating the sensitivity of cells to anticancer therapy were selected [60,61,62,63,64] (Table 2). As resistance biomarkers, miRNAs can be useful in patient stratification and aid in the individualization of therapy.

MicroRNAs are primarily involved in the regulation of the expression of the efflux pumps of the ABC (ATP-binding cassette) transporter family, and thus significantly mediate their participation in the processes of absorption, distribution, and elimination of drugs and the development of resistance to them. By modulating the genes of ABC transporters, microRNAs are involved in the emergence of multidrug resistance, as well as MDR-related mechanisms such as apoptosis, autophagy, drug metabolism, and redox changes [65,66,67]. The P-glycoprotein (P-gp), encoded by the *ABCB1* gene, also known as the multidrug-resistance gene (*MDR1*), is one of the most significant ABC type transporters and is responsible for resistance to a wide range of chemotherapy drugs. It has been experimentally shown that an increased level of P-gp in neoplastic cells may be a consequence of miR-451 and miR-27a overexpression (in the case of MDR and ovarian cancer cell lines) [68,69] or the negative regulatory role of miR-451 (in the case of breast cancer [70], leukemia cell lines [71], and hepatocellular carcinoma [72]). MicroRNAs associated with decreased P-glycoprotein expression have been identified: miR-137 (in the breast cancer cell line MCF-7) [73], miR-145 (in the colon carcinoma cell line Caco-2) [74], miR-200c (in breast cancer cell lines and tissue) [75], miR-223 (in HCC cells resistant to anticancer drugs) [76], miR-298 (in doxorubicin-resistant breast cancer cells [77]) miR-331-5p (in leukemia cell lines) [71], and miR-1253 (in the breast cancer cell line MDA-MB-231) [77]. Regulators responsible for the increase in P-gp expression levels are: miR-27a (in gastric and ovarian cell lines) [70,71], miR-138 (in adriamycin-resistant leukemia cell lines) [78], and miR-451 (in the ovarian cancer cell line A2780) [68]. Regulation of *ABCB1* expression may also be associated with regulation of *ABCB1* activity; consequently, MDR modulation may also be a consequence of the differential expression of miR-381, miR-495 [79], miR-9 [80], miR-122 [81], miR-873 [82], and miR-508-5p [83]. Markers of miRNAs have also been identified for other members of the ABC transporter family, including *ABCG2*/*BCRP* and *ABCC1*/*MRP1* [84].

The consequence of the search for miRNA markers responsible for multidrug resistance in cancer cells is the development of the concept of breaking this resistance. Shang et al. [85] found that controlling the expression of two synergistic miRs, i.e., miR-508-5p and miR-27b, may be of therapeutic benefit in patients with gastric cancer. The study showed that multidrug resistance can be reversed by targeting *ABCB1* and *ZNRD1*. The miR-27b/CCNG1/p53/miR-508-5p axis plays an important role in sensitizing chemoresistant tumors to in vivo and in vitro therapy. Equally promising observations of breaking down the resistance to therapy were made by Bitarte at al. [86] in colorectal cancer stem cells. It has been observed that transfection of the miR-451 precursor leads to a decreased potential for tumorigenicity and self-renewal of colon spheres. The study also showed a decreased level of cyclooxygenase-2 and P-gp, factors that inhibit macrophage migration. On the basis of the obtained results, it was proven that miR-451 can be considered a marker for predicting the response to irinotecan in patients with colon cancer, as a lower expression of miR-451 was characteristic of cells that did not show sensitivity to first-line therapy based on this drug. Pogribny et al. [9] showed the effects of changes in the expression level of miRs on the formation of a cisplatin-resistant phenotype by breast adenocarcinoma cells. The study identified 103 differentially expressed miRs correlated with the MCF-7 cell’s drug resistance. Of these, the largest changes in expression levels were found for miR-146a, miR-10a, miR-221/222, miR-345, miR-200b, and miR-200c. In addition, miR-345 and miR-7 have been shown to suppress drug efflux transporters, including human multidrug resistance-associated protein 1 (*MRP1*) [9]. These observations are interesting because of the dual nature of miR-7, which functions as both a tumor suppressor [7] and an oncomiR [8] for different types of cancer. Hence, it was noted that the inhibition of miR-7 in cancers for which it is an oncomiR can have a general harmful effect by increasing chemoresistance, despite slowing the growth of cancer cells. One of the most widely reported mechanisms for the regulation of epigenetic expression of genes associated with cisplatin resistance through miRNAs is the ovarian cancer model. For this type of cancer, selected marker microRNAs have been associated with a number of cell pathways and processes towards neoplasm, including abnormalities in the course of apoptosis (miR-27a-5p [87], miR-142-5p [88], miR-146a-5p [89], miR-424-3p [90], and miR-454 [91]), cell cycle disturbances and changes in the DNA repair pathway (miR-770-5p [92,93], miR-98-5p [94], and miR-409-3p [95]), abnormalities in signaling pathways (miR-7 [96], miR-106a [97], miR-205-5p [98], and miR-548e [99]), and also disturbances in the distribution of therapeutic particles as well as intensification and detoxification (miR-139-5p [100], miR-194-5p [101], miR-514 [102], and miR-595 [103]). Numerous in vivo and in vitro studies of ovarian tumor cells have demonstrated the possibility of restoring cisplatin sensitivity through targeted microRNA expression, and thus the effectiveness of chemotherapeutic treatment and patient survival. An example is overcoming resistance to this chemotherapeutic drug using miRNAs targeting ABC transporters. A decrease in the level of *ABCB1* expression was made possible by introducing miR-186 [104] or miR-595 [103]. Such modulation towards *ABCB1* suppression had an effect on the inhibition of tumor cell proliferation, metastasis, and drug resistance. A similar effect was observed for miR-130a [105] and miR-873 [82]. A slightly broader spectrum of influence was demonstrated for miR-514, which, through changes in the expression of *ABCA1*, *ABCA10*, and *ABCF2* genes, inhibited the proliferation of ovarian cancer cells and increased sensitivity to cisplatin [102]. The relationship between changes in microRNA expression and the resistance of cancer cells to cisplatin was also demonstrated in the case of many solid and hematological cancers (Table 2).

**Table 2 cells-11-01008-t002:** Marker miRNAs involved in anticancer drug insensitivity mechanisms in human solid tumors.

Drug	Cancer	MicroRNA
Cisplatin	Non-small-cell lung cancer	miR-21 [106,107], miR-107 [108], miR-200c [109], miR-451 [110]
	Lung adenocarcinoma	miR-27a [111], miRNA-378 [112]
	Hepatocellular carcinoma	miR-101 [113], miR-130a [114], miR-182 [115], miR-199a-5p [116]
	Gastric cancer	miR-424 [117], miR-181a-2-3p [118], miR-3180-3p, miR-124-3p [119]
	Ovarian cancer	miR-21 [120,121], miR-125b, miR-133a [122], miR-15, miR-16 [123]
	Osteosarcoma	miR-21 [124], miR-16-5p [125]
	Neuroblastoma	miR-21 [126], miR-141 [127], miR-155 [128]
5-Fluorouracil	Hepatocellular carcinoma	miR-193a-3p [129]
	Colorectal cancer	miR-587 [130], miR-125b-5p [131], miR-375-3p [132], miR-149 [133], miR-135, miR-182 [134], miR-3135b [135]
	Gastric cancer	miR-204 [136], miR-195 [137], miR-30a [138]
	Osteosarcoma	miR-140 [139]
	Lung cancer	miR-27a, miR-27b, miR-134 and miR-582-5p [140]
Methotrexate	Colorectal adenocarcinoma	miR-770-5p [141], miR-24-3p [142], miR-505 [143]
	Lung cancer	miR-200c [144]
	Osteosarcoma	miR-494-3p [145], miR-192 [146]
Doxorubicin	Ovarian cancer	miR-146b-5p, miR-205 and miR-875-3p [147]
	Gastric cancer	miR-494 [148]
	Neuroblastoma	miR-137 [149,150], miR-99b-5p, miR-380-3p, and miR-485-3p [151]
	Breast cancer	miR-200b, miR-17 [152], miR-127, miR-34a, miR-27b, miR-206, miR-21, miR-214, miR-28 and miR-451 [70], miR-200c [153]
Paclitaxel	Ovarian cancer	miR-29a, miR-363, miR-18 and miR-20b [147], miR-130a, miR-30c, miR-335, miR-125b and let-7e [154]
	Prostate cancer	miR-100-5p, miR-200b-3p, miR-34b-3p and miR-375 [155], miR-34a [156,157]
	Breast cancer	miR-21 [158]
	Non-small-cell lung cancer	miR-421 [159], miR-199-5a [160]
Gefitinib	Non-small-cell lung cancer	miR-342-3p [161], miR-506-3p [162], miR-34a [163], miR-564 or miR-658 [164]
Docetaxel	Breast cancer	miRNA-452 [165], miR-34a [166]
	Prostate cancer	miR-181a [167], miR-21 [168,169], miR-134 [170], miR-200 family [171,172]
	Gastric cancer	miR-15b, miR-16 [173]
Oxaliplatin	Colon cancer	miR-137 [174], miR-519d, miR-545, miR-618 and miR-98 [175]
	Colorectal cancer	miR-34a, miR-143, miR-153, miR-27a, miR-218, and miR-520 [176]
	Hepatocellular carcinoma	miR-125b [177]
Topotecan	Ovarian cancer	miR-29a, miR-363, miR-31, miR-18 and miR-20b [147]
	Renal cell carcinoma	miR-21 [178]
	Breast cancer	miR-21 [179]
Fulvestrant	Breast cancer	let-7i, miR-346, miR-638, miR-181a, miR-191, miR-199b, miR-204, miR-211, miR-212, miR-216, miR-328, miR-373, miR-424, miR-768-3p, miR-221/222 [180]
Fludarabine	Leukemia	miR-21 and miR-222 [181], miR-29a, miR-181a, and miR-221 [182], miR-34a [183]
Etoposide	Neuroblastoma	miR-204 [184], miR-520f [185]
	Gastric cancer	miR-15b, miR-16 [173]
	Lung cancer	miR-101 [186]
	Breast cancer	miR-132-3p [187]
Tamoxifen	Breast cancer	miR-221/222 [188,189], miR-449a [190]
Mitoxantron	Breast cancer	miR-155, miR-206 [191], miR-328 [192]

Activation of multidrug resistance is one of the factors of insensitivity to cancer cell therapy. The following are also responsible for the chemoresistance phenotype: abnormalities in autophagic induction (vesicle nucleation, vesicle elongation); changes in the activity of enzymes responsible for drug metabolism (P450 superfamily (CYP) metabolic enzymes); disturbances in the DNA repair pathway, cell cycle, and apoptosis; and changes in the levels of expression of drug targets [84]. Each of these factors has been experimentally shown to be modulated by microRNAs. The effectiveness of anticancer therapy is largely dependent on the proper metabolism of drugs, which is the responsibility of the P450 superfamily, the expression of which is regulated by microRNAs [193]. The results of in vitro and in silico analyses have indicated that as many as 56 CYP enzymes can be regulated by miRNAs [194], directly (by affecting the mRNA of the target cytochrome) or indirectly (by interacting with nuclear receptors (NRs), the constitutive androstane receptor (CAR), or vitamin D receptors) [195]. Cytochromes differ significantly in their extent of regulation, and the length of the mRNA 3′UTR is one of the reasons for this. Enzymes such as *CYP1A1*, *1A2*, *1B1*, *2B6*, and *3A4* are the target of numerous miRNAs. In turn, *CYP2A6*, *2D6*, *2E1*, and *3A5* are regulated by a few specific miRNAs [196]. Among the regulators of the cytochrome superfamily, studies have mentioned miR-214-3p, miR-552, miR-570, and miR-378a-5p [193], and miR-378* for *CYP2E1* [197]; miR-892a for *CYP1A1* [198]; let-7b for *CYP2J2* [199]; and miR-27a/b [195,200], miR-627 [201], miR-122, miR-378a-5p [202], and miR-148a [203] for *CYP3A4*.

One of the better described mechanisms of regulation is the interaction between miR-27b and *CYP1B1* mRNA, which is responsible for the metabolism of a wide range of drugs. Low expression of this microRNA contributes to the overexpression of *CYP1B1* and thus the induction of resistance to a chemotherapeutic agent (e.g., docetaxel in breast, colon, lung, or pancreatic carcinoma cells) [204,205]. These observations prompted attempts to sensitize cancer cells to drugs by reducing the detoxification of drugs metabolized by *CYP1B1* by activating p53-dependent apoptosis [206]. In their experiment, Tsuchiya et al. [205] used transfection with antisense 20-O-methyl oligoribonucleotides (ASO), which acted as an inhibitor of miR-27, and thus demonstrated miR-27-dependent control of *CYP1B1* gene expression in human MCF-7 breast cancer cells.

### 2.2. MicroRNA and Modulation of Drug Activity

MicroRNAs are broad-spectrum molecules. The same miRs can both increase the proliferation of cancer cells but sensitize them to treatment at the same time, resulting in increased overall survival. As one of the first studies, Esquela-Kerscher et al. [207] confirmed their therapeutic potential in cancer using the suppressive properties of microRNA. They experimentally demonstrated that it is possible to reduce tumor weight in lung cancer (in an animal model) by modulating the expression of let-7. Gasparini et al. [208] have shown that miR-155 increases the sensitivity to ionizing radiation therapy in patients with breast cancer. The goal of such therapy is to induce double-stranded DNA breaks in cancer cells. Researchers have described the mechanism by which miR-155 directly suppresses the expression of *RAD51*, a key protein for homologous recombination of DNA, thus blocking the repair of double-stranded DNA breaks and sensitizing triple-negative breast cancer to ionizing radiotherapy. On the basis of in silico analysis and an experiment with colorectal cancer cell lines, Boni et al. [209] investigated the effects of miR-192 and miR-215 on 5-fluoruracil sensitivity. These molecules are modulators of the expression of thymidylate synthase (TS), the expression of which is a predictive biomarker of responses to 5-FU in gastrointestinal tumors. However, on the basis of an analysis of the obtained results, it was found that lowering TS levels with miRNAs does not significantly affect the sensitivity of tumor cells to 5-FU therapy, although the overexpression of miR-192 and miR-215 is associated with a decrease in cell proliferation. Changes in the cell cycle have been associated with p53 status and p21 and p27 activation. As noted, they may result in 5-FU resistance, independent of thymidylate synthase expression. Hirota et al. [140] identified miR-27a, miR-27b, miR-134, and miR-582-5p as regulators of the sensitivity of lung cancer cells to 5-FU. According to their concept, the mechanism of resistance to this chemotherapeutic agent involves post-transcriptional regulation of the expression of dihydropyrimidine dehydrogenase (DPD), which is involved in the metabolism of 5-fluorouracil. The high activity of DPD in cancer cells is an important factor in the efficacy and toxicity of 5-FU therapy. Overexpression of these four miRNAs reduced the *DPD* gene.

Analogous relationships have been described for the sensitivity of cells to gemcitabine therapy for tumor cells. The study by Maftouh et al. [210] showed that induction of miR-211 expression in cells increased the sensitivity to gemcitabine and decreased the expression of its target, ribonucleotide reductase 2 subunits (*RRM2*). The chemical resistance of pancreatic cancer to nucleoside analogs (e.g., gemcitabine) is a result of *RRM2* overexpression. Researchers were able to inhibit the migration and invasion of pancreatic ductal adenocarcinoma cells through forced miR-211 overexpression. The chemosensitivity of neoplastic cells may also be regulated by let-7, which is a negative regulator of *RRM2*. Bhutia et al. [211] described the complicated mechanism of post-transcriptional regulation of *RRM2* and chemotherapy sensitivity by let-7a, as well as the sensitization of PDACs to gemcitabine. These are just a few examples of the regulation of tumor cell responses to anti-cancer drugs available in the scientific literature (Table 2 and Table 3). Most chemotherapeutic agents have attempted to describe the mechanisms of changing the sensitivity profile through the interaction of miRNAs with the mRNA of the target genes involved in drug metabolism or that are effectors.

## 3. MiRNA Delivery Systems

One of the greatest challenges of microRNA therapy is the development of efficient methods of delivery to effector cells. The method must provide both protection against unwanted degradation, and delivery into the cell and uptake without inducing an immunogenic response. Nanoconstructs used for delivery systems must be biocompatible and made of biodegradable materials [261,262]. Due to the small size of the molecule, strategies for delivering microRNA to the cell are similar to those used for interference RNA (siRNA) [227]. The most commonly used carriers are viral and nonviral vectors [263], with viral vectors having lost their importance due to triggering an immune response. Hence, nonviral vectors (e.g., polymeric vectors, lipid-based carriers, and inorganic materials) may be of the greatest importance in anticancer therapy [264,265].

Several routes of introducing the therapeutic construct into the organism have been tested in in vivo studies. Among them, the most frequently chosen are:-Tail vein, e.g., cationic liposomes with miR-29 in lung cancer [214], PEI-PEG with miR-34a in hepatocellular carcinoma [266], and exosomes with miR-145 in lung cancer [267];-Intratumoral, e.g., cationic liposomes with miR-7 in lung cancer [268], polymeric micelles with miR-205 in pancreatic cancer [269], and exosomes with miR-146b in glioma [270];-Intravenous, e.g., carbonate apatite with miR-4711-5p in colon cancer [271], atelocollagen with miR-16 in prostate cancer [272], and exosome-GE11 peptides with let-7 in breast cancer [273];-Subcutaneous, e.g., ionizable liposomes with miR-200c in lung cancer [274], PEI with miR-203 in basal cell carcinoma [275], and atelocollagen with mir-34a in colon cancer [246];-Intraperitoneal, e.g., PEI with miR-145 in colon carcinoma [249] and exosomes with miR-122 in hepatocellular carcinoma [276].

## 4. Therapeutic Approaches Using miRNA

The goal of miRNA-based therapies is to restore the normal function of deregulated cell pathways. There are two possible approaches, including through inhibition of oncogenic microRNA activity (miRNA inhibitors) or by restoring the function of tumor suppressor microRNAs (miRNA mimics). Oncogenic miRNAs can be blocked by using antisense oligonucleotides (ASO), and locked nucleic acids (LNA) such as antimir, anti-mir oligonucleotides (AMO), and antagomirs [4,277]. Another popular strategy for restoring miRNA activity includes the introduction of miRNA mimics or microRNAs coded by expression vectors (Figure 1). The development of an effective therapy requires not only the selection of the correct expression-modulating molecules but also the development of an appropriate cell delivery strategy. Viral and nonviral vectors (polymers and liposomes) and nanoconstructs have been used in the group of carriers. Methods based on nanotechnology are being developed and tested in terms of their potential clinical application in solid tumors. Tumor suppressor microRNAs are the most frequently used in supporting anticancer therapy. Their introduction into the cell reactivates cellular protherapeutic pathways. This approach is known as “miRNA replacement therapy”.

MicroRNAs, as regulators of genes that are important for cancer progression, are increasingly being used in developing new therapeutic concepts in oncology. Analogous to the use of antisense mRNA and RNAi, miRs can be used to regulate the expression of the genes involved in carcinogenesis. High expression of most oncogenes is one of the key factors in the initiation of cancer (Figure 1). According to this assumption, artificial miRNAs are designed to block their expression based on the complementary miRNA properties of their target mRNAs, targeting silencing selected oncogenes. An example is the study of He et al. [27], in which it was confirmed in a mouse model that the induced expression of the miR-17-92 cluster resulted in strong inhibition of *c-myc*-induced apoptosis. The consequence of these changes was intensification of the tumor process.

It has been estimated that such therapy is less toxic compared with other anticancer drugs. For this reason, nanoconstructs containing synthetic antisense oligonucleotides coding complementary sequences to deliver mature oncogenic miRNAs (anti-miRNA oligonucleotides (AMOs)) are being developed [3]. The goal of such therapy is to effectively inactivate the overexpressed miRNAs in cancer cells and, consequently, slow their growth. Clinical trials have confirmed the ability of this class of drugs to significantly suppress the target genes’ expression. Silencing of targeted miRNAs in vivo could be achieved by using antisense oligonucleotides with various nucleic acid analogs involving LNA, AMO, PNAs, or nanoencapsulated PNAs [278,279,280,281]. Many antimiR delivery and targeting strategies have been described (Table 3).

There have been numerous experimental and preclinical attempts to develop therapy protocols. In a mouse model, Krützfeldt et al. [282] showed the possibility of effective inhibition of miRNA activity in various organs via antagomirs in the form of cholesterol-conjugated AMO. Interesting observations were also made by Dickins et al. [283], who found that miR-30 based shRNA (shRNA-miRs) inhibits gene expression under the control of Pol II promoters. Researchers also observed that with Trp53 knockdown using tetracycline-based systems and gene knockdown by the expression of shRNA-miRs (similar to the overexpression of protein-coding cDNAs), it was possible to control tumor growth. Another approach is to use miRNA overexpression techniques, based on transient expression systems, that function as tumor suppressors (e.g., encoded by the let-7 family). This method uses viruses or liposomes, which supply large amounts of miRNA. The construct uses the flanking sequences of pre-miRNA under the control of tissue-specific promoters, thanks to which, it is possible to stimulate and control the endogenous expression of selected molecules in the target cells. Nevertheless, it has been experimentally shown that an immune response may be an obstacle to achieving a therapeutic effect [284,285]. Gokito et al. [286] confirmed the therapeutic efficacy of forced expression of mimic miR-634 in a mouse model. In their study, they used a system of lipid nanoparticles (ionizable lipids: L021-LNP) that they administered to subjects intravenously. After introducing the constructs into the system, they observed the pro-apoptotic effect of the therapeutic agent, resulting in inhibition of pancreatic tumor growth. The disadvantage of this solution was, however, moderate hepatotoxicity. Wang et al. [287] analyzed the effect of miR-16-5p expression in breast cancer cell lines on *ANLN* inhibition. In their model, they used mimic miR-16-5p and si-ANLN, thanks to which, they demonstrated the effect of this marker miRNA on slowing down the proliferation and inhibition of cell migration and invasion. After miR-16-5p overexpression, breast cancer cells were arrested in the G2/M phase. These observations were consistent with those previously published by Magnusson et al. [288], who also demonstrated activation of the apoptotic processes of tumor cells after miR-16-5p overexpression by mimic miRNAs, resulting in subsequent suppression of *ANLN* expression.

The effectiveness of antimiR-based therapies in vivo is hampered by physiological and cellular barriers to delivery to the target cells. Cheng et al. [289] attempted to overcome the barriers of the tumor microenvironment in order to effectively provide therapeutic antisense oligomers. Researchers have developed a platform that targets the acidic tumor microenvironment, evades systemic clearance by the liver, and facilitates cell entry via a non-endocytic pathway. For this purpose, they developed a model targeted therapy, based on a construct containing the peptide nucleic acid (PNA) antimiRs and a peptide with a low pH-induced transmembrane structure (pHLIP) [289]. Researchers first tested platforms containing miR-155 (pHLIP-anti155) delivered to A549 cells and Toledo diffuse large-B cell lymphoma (DLBCL) cells. Nevertheless, they confirmed that the method can be effective for other miRNA molecules (e.g., miR-182, miR-21, and miR-210) and for many other types of cancer cells. The condition is the course of endocytosis, taking the transport properties of pHLIP into account. Brognara et al. [290] tested the biological activity of PNA directed against miR-221 in U251, U373, and T98G human glioma cell lines, using a PNA construct conjugated to an arginine peptide tail.

## 5. Limitations of Replacement Therapy

One of the key challenges in implementing miRNA therapy is the development of clinically cost-effective and effective delivery materials. MicroRNA therapies appear to be effective, especially when mimic molecules are used as endogenous miRNAs to restore tumor suppressor function [264,291]. Nevertheless, the systemic introduction of microRNA mimics carries the risk of integrating their function not only in neoplastic cells but also in properly functioning cells [292,293]. Despite the promising data from in vitro experiments and animal models for breast, intestine, gastric, lung, and hematological neoplasms confirming their therapeutic efficacy, side effects in the form of toxicity and induction of immune and inflammatory responses have also been noticed [294].

Despite the promising results of scientific and preclinical research, attempts to implement therapies with the use of the miRNA nanostructure in oncological clinical trials have been unsuccessful (Table 4). The main reasons are the technical barriers to the effective introduction of therapeutic molecules into the body, especially degradation by nucleases and an unfavorable immune reaction. One of these disadvantages was noted at the stage of Phase I clinical trials in an attempt to introduce miRNA replacement therapy with MRX34, the aim of which was to restore miR-34 expression in cancer patients [295]. Due to the strong immune responses that resulted in the death of four patients, the trials were stopped.

Quite promising results of studies on the introduction of miRNA into clinical practice were obtained by van Zandwijk et al. [296] in a Phase I study, which checked the safety and activity of miR-16-loaded bacterial minicells (TargomiR) in the treatment of patients with recurrent malignant pleural mesothelioma. The nanoconstruct targeted *EGFR* (a mesothelioma overexpression receptor), and the main targets were genes involved in the progression of this cancer, e.g., *BCL2*, *CDK1*, and *JUN*. The use of the miR-16 mimic is a new therapeutic approach for this cancer, especially as palliative chemotherapy is the only available course for the majority of patients [297]. The developed nanoconstruct had to overcome the fibrous nature of the tumor barrier and to protect the nucleic acid from degradation in the peripheral line after intravenous administration. However, the authors failed to confirm the effective delivery of miR16-mimetics to tumor sites in vivo. In addition, many side effects were observed, the most serious of which was increased inflammation. The bacterial origin of the carrier was indicated as the probable cause of the induction of inflammation [296,297,298].

## 6. Conclusions

MiRNom analyses provided insights into the underlying mechanisms of oncogenesis. As strategic regulators of gene expression, small microRNAs are believed to be relatively simple for designing therapeutic agents as compared with antisense oligonucleotides, DNA and mRNA vaccines, or gene therapy vectors. The advantage of miRNAs is that as natural cell components, they should not cause undesirable effects and toxicity. The importance of microRNAs as therapeutic agents in recent years has increased, along with growing knowledge about the changes in cancer miRNAs affecting the response to therapy. It is therefore not surprising that there have been more and more attempts to include miRNAs in therapeutic protocols as part of targeted drugs. The promising results of experimental studies have proven the high effectiveness and low antigenicity of such personalized therapy. However, the clinical use of miRs in cancer therapy primarily requires the identification of specific miRNAs in a particular type of cancer and understanding their mechanisms of action. The next step is to develop a range of therapeutic manipulation methods and a way to deliver miRNA to target cells/tissues that has efficacy after overcoming immune barriers, as well as maintaining their stabilization and continuous activity. There is also no lack of scientific evidence that nanoconstructs containing mimics or antagomirs face barriers related to immunosuppression. Therefore, despite the promising results, the introduction of miRNAs as routine therapy in clinical practice is significantly difficult.

## Figures and Tables

**Figure 1 cells-11-01008-f001:**
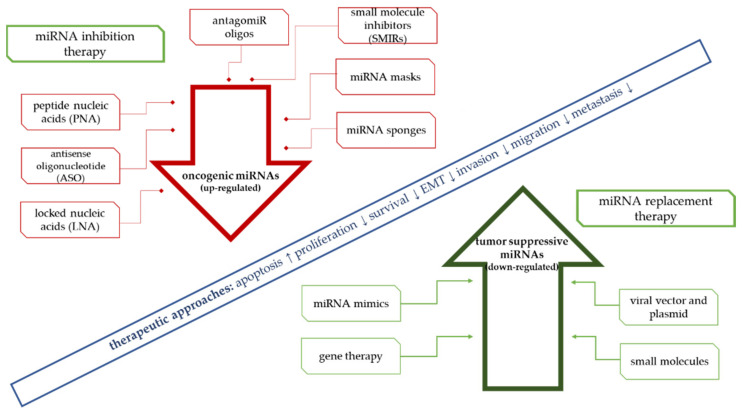
Approaches for miRNA-based therapies.

**Table 1 cells-11-01008-t001:** Classes of miRNAs in cancer.

Classes	Mechanism of Regulation	Examples of Molecules	References
Metastatic miRNAs (metastamiRs)	Significant factors initiating metastasis Regulation of oncogenes, tumor suppressor genes, metastasis genes, cancer stem cell properties, epithelial–mesenchymal transition (EMT), microenvironment, and exosome secretion	miR-7, miR-10b, miR-17/20, miR-19a, miR-34, miR-133a, miR-182, miR-200 family, miR-205	[18,19,20,21]
Oncogenic miRNAs (oncomiRs)	Promoting tumor development by negatively inhibiting tumor suppressor genes Controlling the timing of cell differentiation and proliferation, and cell-cycle exit Regulation of the expression of an oncogene, specifically the *Ras* genes	miR-10b, miR-19a, miR-24, miR-155, miR-181b, let-7 family, miR-17–92 cluster, miR-221/222	[12,13,22,23,24,25,26,27,28,29,30]
Tumor suppressor miRNAs	Inhibition of cancers by regulating oncogenes and/or genes that control cell differentiation or apoptosis Their targets are oncogenes in cell differentiation, cancer invasion, apoptosis, proliferation, and metastasis	miR-35b, miR-145, miR-205, mir-200 family	[5,24,31]

**Table 3 cells-11-01008-t003:** List of microRNAs with therapeutic potential in human cancers.

Cancer Type	MiRNAs	Target Genes	Transfection System	Category	References
Lung cancer	Let-7	*KRAS*	Adenoviral vector	miRNA mimics	Esquela-Kerscher et al. [207]
	Let-7, miR-34a	*KRAS*, *P53*	Neutral lipid-based particles, neutral lipid emulsion	miRNA mimics	Stahlhut and Slack [212], Trang et al. [213]
	miR-29b, 133b	*Mcl-1*, *DNMT3*, *CDK6*	Cationic lipoplex	miRNA mimics	Wu et al. [214], Wu et al. [215]
	miR-34a	*Bcl-2*, *KRAS*	Lentiviral vector neutral lipid	miRNA mimics	Kasinski and Slack [216], Wiggins et al. [217]
	miR-145	*Oct4*, *Sox2*	Cationic polyurethanes, short branch PEI-mediated	miRNA mimics	Chiou et al. [218]
Breast cancer	miR-10b	*Hoxd10*	pcDNA5-CMV-d2eGFP vector	Antagomir	Ma et al. [219]
	miR-19a-3p	*Fra-1*	Nanoparticles	miRNA mimics	Yang et al. [220]
	miR-27a, miR-451	*MDR1*/*P-glycoprotein*	Lipid	miRNA mimics/antagomirs	Zhu et al. [68]
	miR-34a	*E2F3*, *CD44*, and *SIRT1*	T-VISA system (plasmid)	miRNA mimics	Li et al. [221]
	miR-145	*fascin*-1, *c-M**yc*, *SMAD2*/3 and *IGF*-*1R* *p53*, *c-Myc*	Adenoviral vector, lentiviral vector	miRNA mimics	Kim et al. [222] Sachdeva et al. [223]
	miR-326	*MRP-1/ABCC1*	pGL2-control vector	miRNA mimics	Liang et al. [224]
	miR-298, miR-1253	*MDR1/P-glycoprotein* (*P-gp*)	Lipofectamine vector	miRNA mimics	Bao et al. [77]
Hepatocellular carcinoma	Let-7g	c-*Myc*, *p16* (*INK4A*)	Lipid	miRNA mimics	Lan et al. [225]
	miR-21	*PTEN*, *hSulf-1*	Liposomes	miRNA mimics	Bao et al. [226]
	miR-26a	*CCNE1*, *CCNE2*, *CCND2*, and *CDK6*	MSCV-derived retroviral construct	miRNA mimics	Kota et al. [227]
	miR-29	*Bcl*-*2*, *Mcl*-*1*	Lipid	Antagomir	Xiong et al. [228]
	miR-101	*Mcl-1*	Lipid	Antagomir	Su et al. [229]
	miR-124	*HNF4a*	Liposomes	miRNA mimics	Hatziapostolou et al. [230]
	miR-122	*ADAM17*, *ADAM10*, *SRF*, *IGF-1R*	Cationic lipid LNP-DP1 particles, lentiviral vector	miRNA mimics	Hsu et al. [231], Bai et al. [232], Tsai et al. [233]
	miR-143	*NF-kappaB*	PLKO-anti-miR	Antagomir	Zhang et al. [234]
	miR-155	*C/EBPβ*, *FOXP3*	Lactosylated gramicidin-based lipid nanoparticles (Lac-GLN)	Antagomir	Zhang et al. [235]
	miR-199a/b-3p	*PAK4*, *E1A*	Adeno-associated AAV8, oncolytic adenovirus	miRNA mimics	Hou et al. [236], Callegari et al. [237]
Gastric cancer	miR-34a	*Bcl-2*	Lipid and lentivirus	miRNA mimics	Ji et al. [238]
	miR-126	*Crk*	Lipid	miRNA mimics	Feng et al. [239]
	miR-516a-3p	*SULF1*, *WNT*	Atelocollagen	miRNA mimics	Takei et al. [240]
Colon carcinoma	Let-7	*RAS*, *c-Myc*	Cationic liposomes TransIT-TKO	miRNA mimics	Akao et al. [241]
	miR-15a-16-1	*CCNB1*	Cationic liposomes	miRNA mimics	Dai et al. [242]
	p21-targeting miRs	*p53*	Recombinant adenovirus (Ad-p53/miR-p21)	miRNA mimics	Idogawa et al. [243]
	miR-27a	*SGPP1*, *Smad2*	Lipofectamine	miRNA mimics	Bao et al. [244]
	miR-27b	*VEGFC*	Cholesterol conjugate	miRNA mimics	Ye et al. [245]
	miR-34a	*E2F*, *P53*	Atelocollagen	miRNA mimics	Tazawa et al. [246]
	miR-133a	*RFFL*	Lipofectamine	miRNA mimics	Dong et al. [247]
	miR-143, miR-145	*ERK-5*	Cationic liposomes	miRNA mimics	Akao et al. [248]
	miR-145, miR-33a	*c-Myc*, *ERK-5*, *Pim-1*	PLGA/PEI-mediated miRNA vector delivery system	miRNA mimics	Ibrahim et al. [249] Liang et al. [250]
	miR-145	*STAT-1*, *YES*	Lipid	miRNA mimics	Gregersen et al. [251]
	miR-502	*Rab1B*	Oligofectamine	miRNA mimics	Zhai et al. [252]
Acute myeloid leukemia	miR-29b	*SP1*, *CDK6*, *KIT*	Anionic lipopolyplex nanoparticles	miRNA mimics	Huang et al. [253]
Diffuse large B-cell lymphoma	miR-34a	*FoxP1*	Lipid	miRNA mimics	Craig et al. [254]
Neuroblastoma	miR-17-5p	*p21*, *BIM*	Cholesterol-conjugate	Antagomir	Fontana et al. [255]
	miR-34a	*MYCN*	Anti-disialoganglioside GD2-coated nanoparticles	miRNA mimics	Tivnan et al. [256]
	miR-380-5p	*P53*	Lipofectamine	miRNA mimics	Swarbrick et al. [257]
Pancreatic cancer	miR-34a	*Bcl-2*	Lipofectamine	miRNA mimics	Ji et al. [258]
	miR-34a, miR-143/145	*P53*	Plasmid DNA-complexed nanovector	Antagomir	Pramanic et al. [259]
Prostate cancer	let-7a	*E2F2*, *CCND2*	Lipofectamine	miRNA mimics	Dong et al. [260]

**Table 4 cells-11-01008-t004:** Clinical trials of miRNA therapy in oncology (based on https://clinicaltrials.gov, accessed on 10 January 2022).

Therapeutic Agent	Drug Name (Sponsor)	Clinical Trial Number	Phase Status	Cancer
miR-34 mimic	MRX34 (Mirna Therapeutics, Inc.)	NCT01829971	Terminated (Five immune-related serious adverse events) Withdrawn	Primary liver cancer, SCLC, lymphoma, melanoma, multiple myeloma, renal cell carcinoma, NSCLC
miR-34 mimic	MRX34 (Mirna Therapeutics, Inc.)	NCT02862145	Withdrawn (five immune-related serious adverse events in Phase 1 study)	Melanoma
miR-16 mimic	TargomiRs/MesomiR-1 (Asbestos Diseases Research Foundation)	NCT02369198	Completed	Malignant pleural mesothelioma, non-small-cell lung cancer
anti-miR-155	Cobomarsen/MRG-106/Vorinostat (miRagen Therapeutics, Inc.)	NCT03713320 NCT03837457	Terminated (terminated early for business reasons, not due to concerns regarding safety or lack of efficacy.) Terminated (study no longer needed because eligible subjects may receive treatment with cobomarsen in a crossover arm of the SOLAR clinical trial (NCT03713320))	Cutaneous T-cell lymphoma
anti-miR-155	Cobomarsen/MRG-106/Vorinostat (miRagen Therapeutics, Inc.)	NCT02580552	Completed	CTCL, MF, chronic lymphocytic Leukemia (CLL), diffuse large B-cell lymphoma (DVBCL), activated B-cell (ABC) subtype, adult T-cell leukemia/lymphoma (ATLL)

## Data Availability

Not applicable.

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
