# Peer review of "MicroRNA as a Potential Therapeutic Molecule in Cancer"

_cells, 2022, doi:10.3390/cells11061008_

Round 1

Reviewer 1 Report

This study demonstrates a comprehensive review of the therapeutic potential in a variety of human cancers. The ongogenic and tumor suppressive miRNAs are defined and the molecular mechanisms are elucidated. The miRNAs implicated in anti-cancer drug sensitivity are well-identified. The miRNA-targeted clinical trials are also reviewed. Nevertheless, there are a multitude of research articles with regard to the therapeutic role of miRNA in cancers each year. It is surprising that less than a quarter of the references in this manuscript are within the last five years. As far as I'm concerned, the authors raised an excellent scope, yet they must introduce much more recent studies to achieve scientific impact. 

Author Response

Thank you very much for Your thorough analysis and suggestions, according to which we have expanded the scope of the study by the 46 newest articles. We are not able to include too much articles, as it would be beyond the scope of this study, and the review article could be the size of a monograph. We tried to give it a summary context. We realize that this is quite a "hot topic" and the therapeutic role of miRNA molecules is widely commented on in the literature. However, we hope that our concept of presenting the topic will be appreciated for publication.

Reviewer 2 Report

Joanna and colleagues' review study is advanced in the field, well written, and well updated across contexts. There are a few minor points that need to be addressed before the article can be accepted.
Overall English language and style are acceptable but minor spell check is required.
Recently, a study (PMID: 32987354) shows an important role of miR-16 in the induction of autophagy, therefore, I suggest the authors include this study in line 94.

Author Response

Thank you for your constructive and positive comments. As suggested, we added the proposed manuscript to our study. We also followed this lead and wrote a little more about the role of miR-16 in cancer therapy.

Round 2

Reviewer 1 Report

The inclusion of the newest articles is appreciated. As far as I'm concerned, this manuscript is fit to be accepted for publication.